# Concurrent Validity Between EQ-5D and HRQ-6D Measures in Patients with Different Primary Diagnoses

**DOI:** 10.3390/jcm14010064

**Published:** 2024-12-26

**Authors:** Mohamad Adam Bujang, Yoon Khee Hon, Wei Hong Lai, Eileen Pin Pin Yap, Xun Ting Tiong, Selvasingam Ratnasingam, Alex Ren Jye Kim, Masliyana Husin, Yvonne Yih Huan Jee, Nurul Fatma Diyana Ahmad, Cheng Hoon Chew, Clare Hui Hong Tan, Sing Yee Khoo, Fazalena Johari, Alan Yean Yip Fong

**Affiliations:** 1Clinical Research Centre, National Institutes of Health, Sarawak General Hospital, Ministry of Health Malaysia, Kuching 93586, Sarawak, Malaysia; mohamadadambujang1980@gmail.com (M.A.B.); laiweihong@crc.moh.gov.my (W.H.L.); eileenyap.crc@gmail.com (E.P.P.Y.); tiongxt.crc@gmail.com (X.T.T.); khoosingyee@gmail.com (S.Y.K.); 2Institute for Clinical Research, Ministry of Health Malaysia, Shah Alam 40170, Selangor, Malaysia; masliyana.h@moh.gov.my; 3Sarawak General Hospital, Ministry of Health Malaysia, Kuching 93586, Sarawak, Malaysia; selvasingam@gmail.com (S.R.); alexkimsgh@gmail.com (A.R.J.K.); yvonnejee91@hotmail.com (Y.Y.H.J.); clare.tan.hui.hong@gmail.com (C.H.H.T.); 4Heart Center, Sarawak General Hospital, Ministry of Health Malaysia, Kuching 93586, Sarawak, Malaysia; nurulfatmadiyana86@gmail.com (N.F.D.A.); zazy2182@yahoo.com (F.J.); alanfong@crc.gov.my (A.Y.Y.F.); 5Institute for Medical Research, Ministry of Health Malaysia, Shah Alam 40170, Selangor, Malaysia

**Keywords:** concurrent validity, EQ-5D, HRQ-6D, primary diagnoses, health-related quality of life

## Abstract

**Background/Objectives:** The HRQ-6D is a newly developed instrument to measure Health-related quality of life (HRQOL) and EQ-5D is the gold standard for measuring HRQOL. This study aims to test the concurrent validity between EQ-5D and HRQ-6D measures among patients with different primary diagnoses. **Methods**: This cross-sectional study uses two HRQOL measurement instruments, EQ-5D-3L and HRQ-6D. Data collection was performed between January 2023 and May 2023. All the necessary data for this study were collected from actual patients who presented with any one of the four different types of primary diagnoses: heart disease, cancer, depressive disorders, and end-stage renal disease (ESRD). They were recruited from the four specialist clinics that cater to the treatment of each of the four different types of primary diagnoses in a tertiary hospital. **Results**: There were 149 patients who participated in the study wherein 40 (26.8%) of them were ESRD patients, 39 (26.2%) of them were cancer patients, 38 (25.5%) of them were mentally depressed, and the remaining were patients with heart diseases. The domains in HRQ-6D, except for the perception of future health, are significantly associated with domains in EQ-5D-3L after having controlled for patients’ primary diagnoses (i.e., *p* < 0.001). The HRQ-6D replaces the domain “Usual activities” with “Physical energy,” and the association between these two domains is significant (*p* < 0.001). The correlation between the overall HRQ-6D and EQ-VAS scores is also significant (coefficient = 0.445, *p* < 0.001). **Conclusions**: The HRQ-6D is demonstrated to have concurrent validity with EQ-5D. Therefore, clinicians and researchers can use HRQ-6D to measure patient outcomes for interventional and observational studies. (Total word count = 265 words).

## 1. Introduction

Measuring health outcomes using an adequately validated measure is crucial for an accurate assessment of a patient’s health outcomes [1,2,3]. The concept of health-related quality of life (HRQOL) is often used as the patient-reported outcome measure to report the results of a vast majority of clinical research. Researchers can determine whether a patient’s health-related needs are met only if a reliable and valid scale is used to measure the patient’s HRQOL in the studies [3,4,5]. Therefore, a scale for measuring HRQOL should be developed by adhering to a list of detailed and standardized processes [6,7].

EQ-5D was developed by EuroQOL Group, and it aims to measure current and generic HRQOL within 24 h. It has five domains, which are supplemented by the incorporation of a generic visual analog scale (VAS) that measures the current state of an individual’s general health based on a score that ranges from 0 to 100 [8]. The EQ-5D was translated into various languages and has been widely used in clinical research worldwide [9,10,11]. On the other note, the Health-related Quality of Life with Six Domains (HRQ-6D) is a scale that aims to measure current and generic HRQOL within 24 h [12]. HRQ-6D consists of six domains and 12 items, where two items are used to measure each domain. The overall framework of the HRQ-6D scale has also been successfully validated based on both Exploratory Factor Analysis (EFA) and Confirmatory Factor Analysis (CFA). The six domains can then be further categorized into three dimensions: Health (Pain, Physical function, and Psychological symptoms), Body function (Mobility and Self-care), and Perception (Perception of future health), based on an evaluation of their construct validity [12]. A full description of all six domains of HRQ-6D is presented in Table 1.

In addition, the comparison between EQ-5D and HRQ-6D is also presented in Figure 1. The EQ-5D has five domains, while the HRQ-6D has six domains. Overall, the HRQ-6D scale measures a much broader scope of HRQOL by introducing new domains such as “Physical Energy” and “Perception of Future Health”. It also demonstrates improved psychometric performance, as evidenced by various statistical measures of its validity and reliability. Compared to the EQ-5D, the HRQ-6D includes more items, requiring respondents to spend more time filling in all the items of this scale. Each domain in the HRQ-6D was reported to have a minimum value of Cronbach’s alpha of 0.731, and the model fit for the overall framework of the HRQ-6D was also excellent. Since the HRQ-6D has recently been developed and validated, this study aims to take a further step by comparing its psychometric performance against the gold standard EQ-5D among patients with a wide range of different primary diagnoses [12].

By definition, concurrent validity is about how closely a measure is able to match up to an established criterion or gold standard, which can be another measure [7,13]. Our explanation that EQ-5D can be used as the benchmark scale is because: first, HRQ-6D is also used for clinical studies to measure patients’ health outcomes, similar to EQ-5D [14,15,16]. Second, EQ-5D is designed to measure a short-term HRQOL outcome, similar to HRQ-6D, and third, EQ-5D and HRQ-6D have almost the same domains [8,12]. The agreement between EQ-5D and HRQ-6D indicates that the new measure can reliably replicate the gold standard’s results, offering a practical tool for researchers or clinicians. High concurrent validity suggests that the measure is suitable for use in similar contexts as the gold standard, potentially providing a more accessible, cost-effective, or efficient alternative.

## 2. Materials and Methods

This cross-sectional study aims to compare the psychometric properties and clinical utility of HRQ-6D against the conventional standard of EQ-5D-3L. Both scales are regarded as two generic measures of health status among a study population of real patients with different primary diagnoses. Data collection was performed from January 2023 to May 2023. The inclusion criteria for this study were (i) any patients of at least 18 years of age who were currently receiving follow-up treatment in any of the four specialist clinics (i.e., cardiology, oncology, psychiatry, and end-stage renal disease (ESRD) specialist clinics), and (ii) voluntarily agreeing to provide consent to participate in the study. Nevertheless, patients with severe medical conditions, such as advanced organ failure requiring ICU admission at the time of recruitment or experiencing major depressive disorder, would be excluded from this study. Permission to use the EQ-5D-3L Malay version was granted by the EuroQOL Research Foundation. The Malay version of EQ-5D-3L was validated in a previous study [17]. Both instruments were administered to patients from each of the four specialist clinics. The patients were allowed to respond to all the questions independently within 15 min, and then the filled questionnaires would be returned to the researchers for data compilation. The survey was conducted based on a self-administered questionnaire. Finally, the authors were not given access to any information that could identify individual participants during or after data collection.

### 2.1. Ethical and Regulatory Considerations

Each study respondent would be supplied with a participant information sheet that provided adequate information about the purpose of this study before he/she was invited to provide written informed consent for their participation in this study. Only those study respondents who had given prior written informed consent to participate in this study were surveyed. This study adhered to all the relevant guidelines and regulations stipulated by the Medical Research and Ethics Committee (MREC), National Institutes of Health, Ministry of Health Malaysia, ensuring that each study respondent would only have to fill in their responses in the questionnaire provided by this study, without having to retrieve any further information from each individual patient’s medical records. Upon compilation of all the study respondents’ responses in the questionnaires, each of these questionnaires would be anonymized before they were accessed by the relevant personnel for subsequent analysis and interpretation. Ethical approval for this study was granted by the Medical Research and Ethics Committee (MREC), NMRR ID-21-01979-XDL (IIR) on 8 February 2022. This study did not obtain any funding.

### 2.2. Sample Size Planning

The sample size statement of this study follows a guideline introduced in a previous study [18]. This study aims to compare the EQ-5D-3L (which includes three response options for each dimension) and the HRQ-6D measure (which provides an overall score) among patients with four distinct primary diagnoses. A one-way ANOVA was used to determine the sample size, with parameters set at an effect size of 0.30, a power of 0.80, and a significance level of 0.050. The analysis indicated that a minimum of 28 participants per group was required [19]. This calculation was performed using PASS software (PASS 2021 Power Analysis and Sample Size Software, NCSS, LLC, Kaysville, UT, USA, https://www.ncss.com/software/pass/ (accessed on 15 August 2024)). To allow for a potential 10% non-response rate, the required sample size was slightly inflated to 32 participants per group, yielding a total of 128 patients across the four groups.

### 2.3. Statistical Analysis

There were six participants who did not declare their ethnicity. These missing values were imputed according to their religion. In Malaysia, the majority of Malay are Muslim, Chinese are Buddhist, and native Sarawak are Christian. Descriptive analysis was used to describe the socio-demographic profile of the patients and the comparisons made between EQ-5D-3L and HRQ-6D measures in all four different groups of patients. The development of the scoring mechanism of HRQ-6D was described in a previous study [12]. A one-way Analysis of Variance (ANOVA) was applied to evaluate the concurrent validity of EQ-5D-3L and HRQ-6D. A multivariate analysis using General Linear Model Analysis of Covariance (GLM ANCOVA) was applied to assess the concurrent validity between EQ-5D-3L and HRQ-6D after controlling for patients’ primary diagnoses to eliminate any potential sources of confounding. Age groups, gender, and ethnicity were not adjusted in the analysis because these variables were not associated with the status of the EQ-5D and the HRQ-6D scores (*p* > 0.05). A Pearson’s correlation test was used to determine the correlation between the EQ-VAS score and the overall HRQ-6D. All the analyses were performed by using SPSS version 17.0 (SPSS Inc., Released 2008, SPSS Statistics for Windows, Version 17.0, Chicago, IL, USA: SPSS Inc.).

## 3. Results

### 3.1. Profile

There were a total of 149 patients who participated in the study, of whom 40 (26.8%) were patients with ESRD, 39 (26.2%) were cancer patients, 38 (25.5%) were mentally depressed, and the remaining patients were suffering from heart diseases. The majority were females (64.4%), between 18 and 35 years old (30.9%), and Malay (36.5%) (Table 2).

### 3.2. EQ-5D-3L and HRQ-6D

The domains of both EQ-5D-3L and the HRQ-6D are almost identical. However, HRQ-6D has two additional domains, “physical strength” and “perception of future health”, which EQ-5D-3L does not have. The association between the EQ-5D-3L domains and the HRQ-6D domains was found to be statistically significant after controlling for patients’ primary diagnoses (i.e., *p* = 0.001 or *p* < 0.001). The domain “Usual activities” in EQ-5D-3L is associated with the domain “Physical energy” in HRQ-6D (*p* < 0.001). The same analysis was repeated for each subgroup of the four distinct types of primary diagnoses. All domains were significant (*p* < 0.05) except in the heart disease group. Some of the associations were not significant because of the limited sample size in one of the categories in EQ-5D-3L. However, based on the descriptive statistics such as mean (SD), the results show that an excellent HRQOL in EQ-5D-3L (category 1) yielded a higher score based on the HRQ-6D scale in comparison to the other HRQ-6D scores obtained from the EQ-5D-3L categories 2 and 3.

Results show that different categories of health status reported in the EQ-5D yielded significant differences in the mean scores of the HRQ-6D. These results remained significant even after adjusting for differences in primary diagnoses. In other words, when the EQ-5D shows different scores for a broad array of varying categories of health status [which can range from a status of ‘excellent health’ to ‘very poor health’), the results should similarly align with those HRQ-6D scores. This highlights an important observation: when the EQ-5D shows different QoL scores for a wide variety of differing health statuses, the results should similarly align with those HRQ-6D scores (Table 3 and Table 4).

The correlation between EQ-5D VAS score and overall HRQ-6D is presented in Figure 2. The EQ-5D-3L does not have an overall score, but the EQ-VAS score was identified to correlate correspondingly with the overall score of HRQ-6D. This level of correlation was found to be statistically significant (*p* < 0.001) with Pearson’s correlation coefficient = 0.445 (Figure 2). The magnitude of the correlation is moderate and supports evidence of concurrent validity between the EQ-5D VAS score and the overall HRQ-6D. However, other variations, which might possibly be explained by other factors, are not discussed in this paper.

## 4. Discussion

Concurrent validity is one of the most important measurements of validity for a study instrument, and so it is widely regarded as a basis for validity testing during the development and validation of any study instrument [5,20]. Many previous studies involving questionnaire research also successfully adopted EQ-5D as a benchmark tool, during which the evidence of its concurrent validity was also demonstrated [21,22,23,24]. As a pioneer study, the concurrent validity of HRQ-6D is being measured in conjunction with that of EQ-5D, which is regarded as the gold standard for measuring HRQOL. The results of this study have determined that the concurrent validity of HRQ-6D with EQ-5D is excellent. This important finding also demonstrates that further evidence has been accrued to the clinical utility of HRQ-6D as a new scale to measure HRQOL.

The correlation between the EQ-VAS score and the HRQ-6D was also found to be statistically significant, which indicates a moderate level of concurrent validity between EQ-VAS and the overall score of HRQ-6D. However, a major difference arises from the fact that the overall score of HRQ-6D is derived from the scores obtained from six different domains, in contrast to that of EQ-VAS, which is based on a single scale rating ranging from 0 to 100. This means that the overall HRQ-6D score is directly affected by scores obtained from the six domains and therefore researchers will be able to identify which domain(s) or dimension(s) contributed (lower or higher) to the overall HRQOL score. The mechanism of calculating the overall score based on the magnitude of domain scores is well accepted in other QOL scales [25,26,27,28].

Previously, there were numerous debates on the lack of sensitivity of EQ-5D-3L, which then led to the development of EQ-5D-5L [29,30,31,32]. Although the EQ-5D-5L has a greater variety of options, it still holds a similar concept to EQ-5D-3L, whereby each domain is represented by only one item. The design of HRQ-6D is unique because it allows each domain to be represented by two items, which also confers some distinct advantages to the scale of HRQ-6D in comparison to that of EQ-5D. This is because the content of each domain in HRQ-6D has allowed a much broader scope of depiction along with a more precise delineation by including a greater level of detail than that in EQ-5D-3L or EQ-5D-5L. This approach enables other tests, such as Cronbach’s alpha (for assessing internal consistency), to be performed within each domain in HRQ-6D [33].

Another major advantage of the HRQ-6D is that this new scale has incorporated the measures of both “Physical energy” and “Perception of future health”, which are absent in EQ-5D. Physical energy or strength is a crucial health indicator for HRQOL, as described in most other generic QOL scales [25,26]. In a previous study, the inclusion of an additional item, ‘fatigue’, to the EQ-5D was found to improve the performance criteria of an HRQOL measurement instrument in the population, which means that the inclusion of ‘fatigue’ in EQ-5D greatly enhances its relevance in the evaluation of HRQOL of people with current health issues [34]. In addition, studies have shown a strong correlation between HRQOL and fatigue [35,36,37]. Thus, the HRQ-6D has now incorporated a new domain that assesses the level of physical energy of a person. Perception of future health is also important for addressing the level of confidence patients have regarding how their health conditions will progress as time goes by [10,38,39]. Historically, most patients with type 1 diabetes mellitus had a bad perception of their future health before insulin treatment was introduced. However, when insulin injection was introduced to these patients, this chronic disease became more manageable, and most importantly, these patients experienced far better HRQOL [40,41].

The HRQ-6D scale has significant implications for clinical practice by guiding treatment decisions and patient management strategies. Identifying strengths and weaknesses across six domains enables personalized treatment plans, such as prioritizing rehabilitation for low physical function scores or integrating mental health support for emotional well-being concerns [12]. The scale facilitates shared decision-making, aligns care with patient preferences, and supports resource allocation by targeting areas needing improvement. It also aids in monitoring progress, allowing dynamic adjustments to management strategies, and serves as a tool for preventive care, identifying risks early to prevent chronic conditions [33]. Additionally, HRQ-6D promotes holistic care by encouraging multidisciplinary approaches to address specific domains comprehensively. Its use extends to healthcare policy and clinical trials, assessing treatment efficacy beyond traditional outcomes while fostering patient-centered care that enhances satisfaction and adherence to treatment [12,33].

The HRQ-6D measures a critical human asset, which is health. Based on this, the recommendation is to use it as a reliable instrument for assessing an individual’s health-related quality of life and for promoting positive changes and improvements in their quality of life during treatment and clinical interventions. To further enhance the applicability and versatility of the HRQ-6D, it is recommended that future studies utilize it to determine HRQOL among diverse populations with different disease conditions and incorporate its use in performing cost-effectiveness analyses.

### Limitations of This Study

All patients recruited as respondents for this study were currently in stable condition despite being afflicted by severe chronic diseases such as cancer and ESRD. This study focused on recruiting stable patients primarily because severely ill patients would be unable to complete the questionnaire. Additionally, it would be unethical to persuade them to do so. Therefore, this study might not be able to elicit responses from patients who are in a less stable condition or whose conditions are deteriorating rapidly. Thus, the results should be interpreted with caution, taking this limitation into account. Correspondingly, the sensitivity and responsiveness of the mean score of HRQ-6D to assess the overall performance of HRQOL and the psychometric performance of all its domains or dimensions could be affected by such limitations as well. This study used EQ-5D-3L as a benchmark tool for comparison purposes despite there being much published criticism directed at EQ-5D-3L [29,30,31,32]. Future studies can, therefore, be proposed to test the level of concurrent validity between EQ-5D-5L and HRQ-6D.

## 5. Conclusions

In conclusion, this study showed a high level of concurrent validity between EQ-5D and HRQ-6D. Therefore, the HRQ-6D can be used to measure patients’ reported outcomes in both interventional and observational studies.

## Figures and Tables

**Figure 1 jcm-14-00064-f001:**
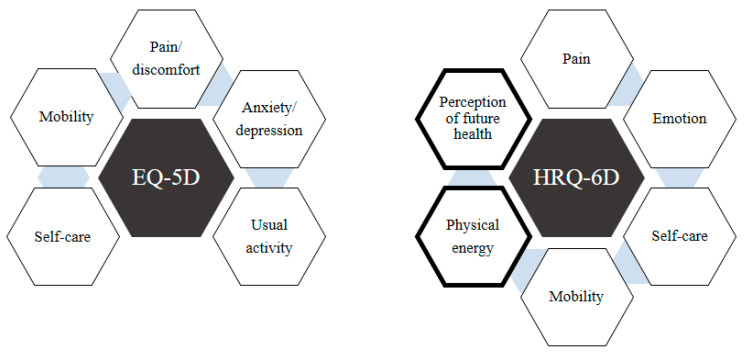
Comparison of EQ-5D versus HRQ-6D.

**Figure 2 jcm-14-00064-f002:**
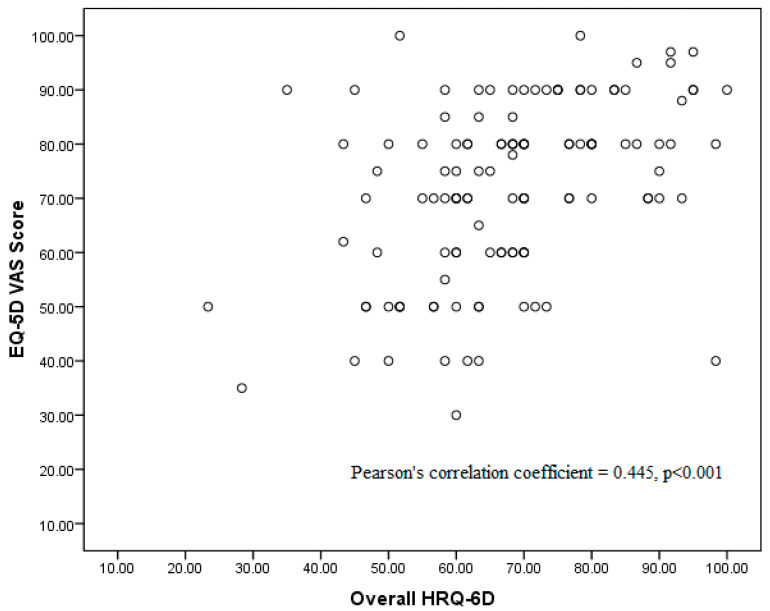
Correlation between EQ-5D VAS score and overall HRQ-6D.

**Table 1 jcm-14-00064-t001:** Description of HRQ-6D.

No.	Domains and Their Description
1	Pain Unpleasant physical sensation in any part of the body caused by illness or injury
2	Physical energy The energy to do physical activities
3	Self-care The ability of a person to perform self-care activities independently
4	Mobility The ability to move on his/her own
5	Emotion The existence of unpleasant emotions due to experiencing unfavorable and/or unexpected conditions
6	Perception of future health A strong belief that something will happen (good or bad) related to his/her health condition

**Table 2 jcm-14-00064-t002:** Basic demographic profiles and primary diagnoses of patients.

Profile/Clinical	Category	*n*	%
Gender	Male	53	35.6
	Female	96	64.4
Age group	18–35	46	30.9
	36–40	21	14.1
	41–50	30	20.1
	51–60	38	25.5
	More than 60	14	9.4
Ethnicity	Malay	63	42.3
	Iban	22	14.8
	Bidayuh	22	14.8
	Other Sarawakian	2	1.3
	Chinese	34	22.8
	Others	6	4.0
Diagnosis	ESRD	40	26.8
	Cancer	39	26.2
	Depressive disorders	38	25.5
	Heart disease	32	21.5

Note: ESRD refers to end-stage renal disease.

**Table 3 jcm-14-00064-t003:** Association between EQ-5D status and HRQ-6D score for the first three domains.

Domains		HRQ-6D Score				
	EQ-5D Status	*n*	Mean	CI	SD	*p*-Value
Pain/discomfort versus Pain	1	89	72.0	68.2, 75.9	18.3	<0.001 ^a^
	2	56	51.6	47.2, 56.0	16.4	
ESRD	1	28	74.6	67.2, 82.1	19.1	0.001
	2	12	54.2	48.4, 59.9	9.0	
Cancer	1	17	75.3	68.2, 82.4	13.7	<0.001
	2	19	53.7	45.2, 62.2	17.7	
Dep. Disorders	1	21	69.5	59.3, 79.8	22.5	0.001
	2	16	46.3	38.5, 54.0	14.5	
Heart	1	23	68.7	61.8, 75.6	16.0	0.041
	2	9	53.3	35.3, 71.4	23.5	
Anxiety/Depression versus Emotion	1	90	79.6	75.5, 83.6	19.3	<0.001 ^a^
	2	41	53.2	47.5, 58.8	17.9	
	3	5	22.0	16.4, 27.6	4.5	
ESRD	1	32	82.8	75.9, 89.7	19.0	0.003
	2	7	62.9	50.1, 75.6	13.8	
	3	1	30.0	nil	nil	
Cancer	1	22	84.5	78.3, 90.8	14.1	0.002
	2	9	64.4	52.2, 76.7	15.9	
Dep. Disorders	1	11	53.6	43.1, 64.2	15.7	0.001
	2	22	43.2	37.5, 48.9	12.9	
	3	4	20.0	nil	0.0	
Heart	1	25	82.4	75.4, 89.4	16.9	0.265
	2	3	70.0	4.3, 100.0	26.5	
Mobility versus Mobility	1	126	80.4	77.5, 83.3	16.6	<0.001 ^a^
	2	17	60.6	52.4, 68.8	16.0	
ESRD	1	34	78.5	73.2, 83.8	15.2	0.005
	2	6	56.7	32.1, 81.2	23.4	
Cancer	1	33	80.9	76.0, 85.8	13.8	0.145
	2	1	60.0	nil	nil	
Dep. Disorders	1	35	83.4	76.5, 90.4	20.3	0.173
	2	3	66.7	28.7, 100.0	15.3	
Heart	1	24	77.9	71.1, 84.7	16.1	0.017
	2	7	61.4	51.5, 71.3	10.7	

Note: ^a ^
*p*-values were derived after controlling for patients’ primary diagnoses in the analysis based on the GLM ANCOVA test. The rest of the *p*-values were derived based on the one-way ANOVA test. The total number of patients in each group (primary diagnosis) did not match the figures in Table 2 due to the presence of missing values. ESRD refers to end-stage renal disease. Dep. disorders refers to depressive disorders.

**Table 4 jcm-14-00064-t004:** Association between EQ-5D status and HRQ-6D score for the remaining domains.

Domains		HRQ-6D Score				
	EQ-5D Status	*n*	Mean	CI	SD	*p*-Value
Self-care versus Self-care	1	136	79.4	76.2, 82.6	18.9	0.001 ^a^
	2	9	55.6	51.5, 59.6	5.3	
ESRD	1	36	78.1	71.1, 85.1	20.7	0.057
	2	4	57.5	49.5, 65.5	5.0	
Cancer	1	36	84.2	78.9, 89.4	15.6	nil
	2	nil	nil	nil	nil	
Dep. Disorders	1	35	76.3	69.0, 83.6	21.3	0.094
	2	2	50.0	Nil	0.0	
Heart	1	29	79.0	72.5, 85.4	17.0	0.033
	2	3	56.7	42.3, 71.0	5.8	
Usual activities versus Physical strength	1	104	66.3	62.3, 70.3	20.6	<0.001 ^a^
	2	39	50.8	44.9, 56.6	18.1	
	3	3	33.3	0.0, 90.7	23.1	
ESRD	1	30	72.3	64.7, 80.0	20.5	0.036
	2	9	52.2	39.0, 65.4	17.2	
	3	1	60.0	nil	nil	
Cancer	1	27	71.9	65.5, 78.3	16.2	0.001
	2	9	50.0	39.8, 60.2	13.2	
Dep. disorders	1	25	52.8	43.8, 61.8	21.9	0.049
	2	11	41.8	31.9, 51.7	14.7	
	3	2	20.0	nil	0.0	
Heart	1	22	66.8	58.9, 74.7	17.8	0.368
	2	10	60.0	43.5, 76.5	23.1	
Perception of future health		149	57.1	53.2, 60.9	23.8	0.662

ESRD		40	55.0	47.7, 62.3	22.9	
Cancer		37	57.8	49.3, 66.4	25.7	
Dep. disorders		38	55.0	46.5, 63.5	25.9	
Heart		31	61.3	54.0, 68.6	20.0	

Note: ^a ^
*p*-values were derived after controlling for patients’ primary diagnoses in the analysis based on the GLM ANCOVA test. The rest of the *p*-values were derived based on the one-way ANOVA test. The total number of patients in each group (primary diagnosis) did not match the figures in Table 2 due to the presence of missing values. ESRD refers to end-stage renal disease. Dep. disorders refers to depressive disorders.

## Data Availability

Data cannot be shared publicly because of privacy and confidentiality concerns. Anonymized data can be made available by making a formal request from the Director General of Health, Ministry of Health, Malaysia for an official approval to provide authorized access to such data. This formal request can be made via the corresponding author.

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
