# Peer review of "Concurrent Validity Between EQ-5D and HRQ-6D Measures in Patients with Different Primary Diagnoses"

_jcm, 2024, doi:10.3390/jcm14010064_

Round 1
Reviewer 1 Report
Comments and Suggestions for Authors
In the abstract, consider replace “One hundred and forty-nine” with just “149”,also in line 30, please replace the word “excellence”, which would not be consistent with scientific expression.
In the introduction, please consider using shorter sentences for readability. Also please highlight the difference of EQ-5D and HRQ-6D.
In the result section, from line 158-159, “The majority of the domains were significant (p<0.05).”should be depicted more clearly.
The shortcomings of HRQ-6D should also be discussed.
Comments on the Quality of English Languagethere are some typos, the writing should be improved.
Reviewer 2 Report
Comments and Suggestions for Authors
This is an interesting paper, the statistical methodology is correct but before it can be published several changes should be done.
1. Table 2, is not mentioned in the text, it should be mentioned.
2. Table 3 is not understood., The column headers are identical, but the results are different. Remember that a reader should be able to understand a table, without a reading the main body of the paper. As mentioned before Table 3 in the paper presents a confusing structure because the column headers are indeed identical in the left and right halves
3. What does it mean in Table 3 in the EQ-5D column 1.0, 2.0, 3.0? The numbers (1.0, 2.0, 3.0) are not clearly explained in the table or in its legend, which could easily confuse readers. Remember that a reader should be able to understand a table, without a reading the main body of the paper
4. section 2.2. Sample Size Planning, Section 2.2, Sample Size Planning the current text is somewhat difficult to read and could benefit from greater clarity and conciseness. It could be somethin like this “Based on one-way ANOVA with an effect size of XXX and power of yyy and and 95% confidence, we calculated a minimum sample of WWW participants per group. To allow for a 10% non-response rate, we increased this to RRR participants per group, totaling LLLL patients. This calculation was done using PASS software PASS 2020 Power Analysis and Sample Size software (2020). NCSS, LLC. Kaysville, Utah, USA, ncss.com/software/pass). The sample size was set to detect differences between groups without adjustment for covariates used in later analyses, as subgroup comparisons were exploratory.” Don’t copy my test use your own words, to mantain the coherence of the paper.
Reviewer 3 Report
Comments and Suggestions for Authors
Dear Authors:
The subject is interesting; however, there are concerns about structure and methodology that should be explained and clarified.
Comments:
Abstract:
1. Although the background and purpose have been presented, in the section of the abstract the broader conceptual significance of the findings on the measurement of most preferred HRQOL presents ambiguity.
2. Statistical reporting, such as confidence intervals or effect sizes, should have been given in this results section to show a stronger and more robust presentation of the findings.
3. It appears as if the conclusion could have been crafted more forcefully, emphasizing implications for future research or clinical practice, rather than simply reiterating that the HRQ-6D had been validated.
Main text:
Introduction:
4. The introduction can be more concise; some sentences are too long and do not neatly fit into the major point being put across. For example, sentence 40 could have been shortened into a simpler sentence for better readability.
5. Some of the phrases used could be less wordy; for instance, "the aim of this study shall be to compare" could be shortened to "this study compares," which is more straightforward and closer to the style guidelines in scientific writing.
6. Although the introduction cites a number of studies, it would have been even better with more recent citations regarding the relevance and clinical research uses of HRQOL measures. This would go a long way towards supporting the necessity argument for the validation of the HRQ-6D.
7. Although introduction refers to some literature, it lacks any proper literature review about any previous research which was done comparing measures of HRQOL. It should have described what gaps exist in the current literature and how this study will fill those gaps.
8. Regarding the sentence about psychometric performance in HRQ-6D, lines 65-66 do not present specific statistical justifications. It would be much better if there were specific metrics provided, for instance, coefficients of reliability or indices of validity.
9. The definition of concurrent validity on lines 70-71, explains concurrent validity as an agreement with a gold standard, but doesn't take that further in terms of explaining its implications. Please add more information.
10. The transition between paragraphs would still need attention to ensure that the logical development of ideas flows; for instance, the discussion of instrument differences could be better linked to the justification for the study.
11. It is confusing to use both "Health-related Quality of Life" and "HRQOL." in the introduction. It is better to use one term.
12. Is this study novel, there are other studies comparing EQ-5D and HRQ-6D. Please, highlight the area of the novelty of this article. In introduction and discussion.
Material and method:
13. Some of the sentences are quite long, please, shorten them for easier reading. For instance, line 85 could be rewritten as, "This cross-sectional study compares the psychometric properties and clinical utility of HRQ-6D and EQ-5D-3L among patients with various primary diagnoses."
14. Whereas the inclusion criteria are well-defined, the exclusion criteria would be better delineated. For example, other than stating "too sick" or "unstable mental condition, provide specific clinical definition or conditions that will merit exclusion.
15. This can also be briefly justified in terms of the period for which data collection was made from January 2023 to May 2023. For example, this could be why the period was chosen, or if it falls in line with any specific clinical consideration.
16. The section of statistical methods is rather technical (lines 136-145). A short explanation of why each method has been chosen and what is expected to be achieved would help readers not directly familiar with such analyses. Also, the statement that missing data will not be imputed may raise concerns about the robustness of the results and should be explained.
17. While the sample size calculation is well elaborated, a better explanation of the effect size and its relation to the study should be provided. Additionally, it would be of great importance to state any similarities this sample size may have with others already conducted on the same topic.
18. The section mentions that both instruments were administered but does not provide information on how they were administered, such as self-administered or interviewer-administered. Its inclusion will help give context to how data are collected.
19. Mention of SPSS software is good; however, it does not mention what version was used to conduct the analyses. At least the exact version should be mentioned here, as was done in line 146 for reproducibility.
20. The analysis plans indicate control for primary diagnoses but do not mention other potential confounders that could affect scores on HRQOL, including demographic factors such as age, gender, and socioeconomic status. Discussion of these would enhance the validity of the study.
Results:
21. It would be very relevant if this section could mention the background of why these particular tests were used; the p-values were derived from GLM ANCOVA and one-way ANOVA tests. A short justification of using those statistical tests would greatly help in making this clear.
22. The note at the bottom of Table 3 states that due to missing values, the sum of the number of patients in each group did not equal Table 2. This would be helpful if one specified the number of patients with missing values and for what reason they were omitted because it would add to the meaning of interpretation of the results.
23. The results for the correlation between EQ-VAS and HRQ-6D are stated, but their elaboration does not reach full development. More context to what is implied from having a Pearson's correlation coefficient of 0.445 would be good to give. The discussion of this level of correlation in terms of clinical relevance would give more weight to the findings.
24. The results only mention that the analysis was repeated for each subgroup of primary diagnosis, but it does not report any specifics or insights from these subgroup analyses. This would offer a fuller picture in summarizing or showing key findings from these analyses.
25. There is little interpretation about what the significant differences in HRQ-6D scores across the EQ-5D categories clinically mean. It would be of more interest, therefore adding value to results, how such findings may affect clinical practice or management of patients clinically from the results.
Discussion:
26. While limitations are identified, they have not been fully discussed. For instance, a discussion on how only stable patients may impact the generalizability; there might be some mention of what this could mean for the interpretation of the HRQ-6D scores within a wide patient population.
27. There is some repetition in certain phrases and ideas that do not contribute anything new. For example, the explanation that HRQ-6D is calculable based on multiple domains could have been summarized to avoid repetition.
28. The term "concurrent validity" appears in a number of places; the discussion would be even stronger with a subtler definition or explanation of what this actually means in practical terms for the HRQ-6D scale at least for readers with modest experience with psychometric concepts.
29. The implications for clinical practice and policy are inadequately discussed. A discussion of how the HRQ-6D might influence a treatment decision or health policy might, therefore, strengthen the rationale for its use.
30. The recommendations in general are not very specific; mere short discussion of future studies. For instance, it might have indicated which populations/settings where HRQ-6D can be tested would provide clearer guidance for future research.
31. The discussion is done by mentioning most of the strengths of the HRQ-6D; however, no weaknesses are pending, nor is there any contradictory evidence from other studies. A balanced view to include criticisms or limitations of the HRQ-6D would add strength to the discussion.
32. The conclusion is repetitive, stating what has already been said, and does not have a deeper impact. It needs to be rewritten to briefly summarize the main findings and implications. Second, it could focus on the possible impact of HRQ-6D on future research and clinical practice.
Comments on the Quality of English LanguagePlease shorten the long sentences. I've provided some examples in the recommendations.
Round 2
Reviewer 2 Report
Comments and Suggestions for Authors
Der authors your changes have improved de manuscript.
Author Response
There are no reviewer comments for the authors to respond to.
Reviewer 3 Report
Comments and Suggestions for Authors
The article is significantly improved compared to your previous suggestions. However, addressing the points mentioned below could further enhance its quality and impact. If the authors implement these suggestions, the article will be even stronger.
1. Although the authors themselves said that effect sizes were not necessary, one might suggest that the confidence intervals of the key results be included. This can be added to reinforce the strength of the statistical reporting.
2. While the introduction now contains recent citations, it would benefit from a broader overview of the existing literature on measures of HRQOL. This would set up the argument for why this study is needed.
3. The discussion section could expand on the implications of recruiting only stable patients. Recognizing how this might impact generalizability would be appropriate.
4. Make specific recommendations for future research, such as what populations or settings might be useful to test HRQ-6D in.
5. The implications for clinical practice could be expanded. Discussing how HRQ-6D might influence treatment decisions or patient management strategies would add depth to the discussion.
